# Relationships between maximum tongue pressure and second formant transition in speakers with different types of dysarthria

**Toshiaki Tamura** [1,2]*, **Yasuhiro Tanaka**[3], **Yoshihiro Watanabe**[4], **Katsuro Sato**[1,2]

**1** Department of Speech, Language, and Hearing Sciences, Niigata University of Health and Welfare, Niigata city, Niigata, Japan, **2** Major in Health and Welfare, Niigata University of Health and Welfare Graduate School, Niigata city, Niigata, Japan, **3** Faculty of Psychological and Physical Science, Aichi Gakuin University, Nisshin city, Aichi, Japan, **4** Department of Rehabilitation, Uonuma Kikan Hospital, Uonuma city, Niigata, Japan

* toshiaki-tamura@nuhw.ac.jp

**Data Availability Statement:** All relevant data are within the paper and its Supporting Information files.

## Abstract

The effects of muscle weakness on speech are currently not fully known. We investigated the relationships between maximum tongue pressure and second formant transition in adults with different types of dysarthria. It focused on the slope in the second formant transition because it reflects the tongue velocity during articulation. Sixty-three Japanese speakers with dysarthria (median age, 68 years; interquartile range, 58–77 years; 44 men and 19 women) admitted to acute and convalescent hospitals were included. Thirty neurologically normal speakers aged 19–85 years (median age, 22 years; interquartile range, 21.0–23.8 years; 14 men and 16 women) were also included. The relationship between the maximum tongue pressure and speech function was evaluated using correlation analysis in the dysarthria group. Speech intelligibility, the oral diadochokinesis rate, and the second formant slope were based on the impaired speech index. More than half of the speakers had mild to moderate dysarthria. Speakers with dysarthria showed significantly lower maximum tongue pressure, speech intelligibility, oral diadochokinesis rate, and second formant slope than neurologically normal speakers. Only the second formant slope was significantly correlated with the maximum tongue pressure ($r = 0.368$, $p = 0.003$). The relationship between the second formant slope and maximum tongue pressure showed a similar correlation in the analysis of subgroups divided by sex. The oral diadochokinesis rate, which is related to the speed of articulation, is affected by voice on/off, mandibular opening/closing, and range of motion. In contrast, the second formant slope was less affected by these factors. These results suggest that the maximum isometric tongue strength is associated with tongue movement speed during articulation.

## Introduction

Dysarthria is a neurological speech disturbance characterized by abnormalities in muscle strength, steadiness, tone, speed, range of motion, and/or accuracy of control of speech organs (e.g., tongue, lips, and larynx) for speech production [1]. Among these multidimensional

**Funding:** This work was supported by the Japan Society for the Promotion of Science Grants-in-Aid for Scientific Research (KAKENHI) (https://www.jsps.go.jp/j-grantsinaid/) under Grant number JP20K19324. The funders had no role in the study design, data collection and analysis, decision to publish, and prepared of the manuscript.

**Competing interests:** The authors have declared that no competing interests exist.

motor disorders, muscle weakness is associated with decreased exercise speed, which is speculated to be associated with slow speech [2]. This hypothesis has received mixed support depending on the correspondence between performance of the syllable repetition task, called oral diadochokinesis (oral-DDK) or alternating motion rate, and muscle weakness [3–5]. However, current reviews have reported no significant relationship between tongue strength and speech-related indicators, such as speech intelligibility, articulation rate, and oral-DDK rate [6–8].

Low levels of orofacial muscle strength are required to generate utterances. The orofacial muscle strength required for normal speech is at most 10%–20% of the maximum muscle strength [9–12]. In speakers with atrophic lateral sclerosis with orofacial muscle weakness (i.e., bulbar paralysis), the ratio of tongue-to-palatal contact pressure during speech to maximum isometric tongue muscle strength is 2%–8%. This ratio is not significantly different from that of healthy participants [10], suggesting that the maximum muscle strength and tongue-palate contact pressure during speech decrease proportionally. Additionally, speakers with dysarthria in whom tongue muscle strength is lower than the lower limit of normal speakers have moderate-to-severely reduced articulatory precision and overall severity (including speech intelligibility and naturalness) [13]. In a previous cross-sectional study [13], dysarthria speakers (n = 8) with severe anterior tongue elevation muscle strength had an oral-DDK rate of <5.8 syllable/s for the syllable /tʌ/. In contrast, 44.6% of the remaining speakers with dysarthria had an oral-DDK rate of >5.8 syllables/s. These findings suggest that a severe decrease in orofacial muscle strength adversely affects speech intelligibility.

Abnormal articulation is the main cause of poor speech [14]. In particular, the tongue (among the articulatory organs) has a strong effect on articulation. Elevation strength of the anterior tongue is correlated well with audibly acquired articulatory precision compared with speech intelligibility [4,13]. However, since audibly articulatory precision and speech intelligibility are qualitative assessments, they affect the distribution of the data (ceiling or floor effect) [13]. A second formant (F2) slope is a quantitative evaluation of articulation. It is a quantitative acoustic measurement based on connected speech (word or sentence level). The F2 slope changes almost in response to the back and forth movements of the tongue [15], and frequency trajectories, such as diphthongs, move up and down relatively rapidly. Therefore, the F2 slope is speculated to reflect the movement speed of the tongue during articulation. A correlation exists between perceptually measured vowel accuracy and the F2 slope [16]. In addition, the F2 slope of the F2 transition is also correlated with speech intelligibility [17–21]. The clear explanation for the decrease in the F2 slope in speakers with dysarthria is the relatively slow changes in tongue shape [22]. Specifically, the back-and-forth motion of the tongue during articulation is slower and/or the range of movement is narrower, resulting in a longer and thinner change in the F2 movement.

The hypothesis that the tongue strength affects the speech speed does not appear to be supported by the weak correlation between the tongue strength and the articulation and oral-DDK rates [13]. However, these rates quantify how fast the syllables are generated, and the accuracy of articulation is not considered much. In addition, there is a trade-off between articulation accuracy and speed [23]. The index for measuring the rate of syllable generation varies from person to person in terms of the actual movement speed of the tongue during articulation (the relationship between movement range and required time). A study [11] investigated the correlation between the range of motion of the oral articulator during articulation and tongue strength. Speakers with oculopharyngeal muscular dystrophy (n = 12) showed no correlation between vowel space area or vowel F2 range and tongue muscle strength. This result may be due to the fact that the range of motion of the oral articulator can be compensated for by

slowing down the speech speed. Given the above limitations, additional research is needed on the relationship between tongue strength and slowed tongue movement during speech.

This study aimed to further elucidate the relationship between tongue muscle weakness and dysarthria in an adult multidisciplinary group. Speech-related indicators have been extended from the well-studied oral-DDK rate and speech intelligibility to include the F2 slope. As mentioned above, the F2 slope directly measures tongue movement during articulation from the viewpoint of range and speed. Furthermore, the anterior tongue elevation and anterior tongue consonant tasks are related [4,13,24]. Our interest in the relationship between tongue strength and the F2 slope was also motivated by the specificity of this site. This study may provide suggestions for adaptation and efficacy verification of strength training for dysarthria.

## Materials and methods

### Participants

This cross-sectional study population included speakers with dysarthria who were admitted to acute and convalescent hospitals between September 2017 and June 2020 and were continuously evaluated by speech-language-hearing therapists (SLHTs). The eligibility criteria were as follows: 1) request made for speech rehabilitation from a doctor; 2) Japanese as the first language; 3) absence of severe cognitive impairment or psychiatric disorders that may hinder speech assessment, 4) no complications of respiratory function that may affect speech, such as pneumonia and asthma; and 5) no dentition defects that affect the production of lingual–alveolar consonants or tongue pressure measurement. Background information, such as age, sex, height, weight, albumin, and diagnosis were obtained from the medical records of the participants. Body mass index was calculated as weight (kg) divided by height in meters squared ($m^2$). These factors were considered to account for the possible effects on tongue muscle strength and speech caused by factors other than the primary disease causing dysarthria. Classification of the dysarthria type was diagnosed by SLHTs using the Mayo Clinic classification system [1].

Thirty neurologically normal speakers (14 men and 16 women) aged 19–85 years (median, 22 years; interquartile range, 21.0–23.8 years) were also included. This group comprised 25 participants reported in a previous study [25] and five selected in their preliminary experiments. It also included one participant with esophageal cancer (76 years old, male) and one with a trochanteric fracture of the femur (85 years old, female). These participants comprised the control group for the tongue pressure and speech-related indicators in this study. Their heights, weights, and albumin levels were not included.

### Tongue strength

A balloon-type tongue pressure measurement device (TPM-01; JMS Co. Ltd, Hiroshima, Japan) was used to measure tongue strength. Maximum tongue pressure (MTP) measurements were performed by six SLHTs, including the trained author (WY). The reproducibility and reliability of this device have been validated in a previous study [26]. The measured values were calculated according to a previously established methodology [26–29]. The TPM-01 comprises a disposable probe, an injection tube as a connector, and a hard ring (bite block; length, 8.5 mm; thickness, 0.5 mm; diameter, 6.0 mm) device (Fig 1).

The participants were instructed to put the balloon in their oral cavity in a sitting position. They held the probe at the midpoint of the central tooth. The participants were asked to maintain this position while the measurer adjusted the probe and confirmed the correct position. Measurements were performed thrice with 1-min rest and one preliminary exercise. The maximum value of the three measurements was defined as the MTP in kilopascals (kPa). In this

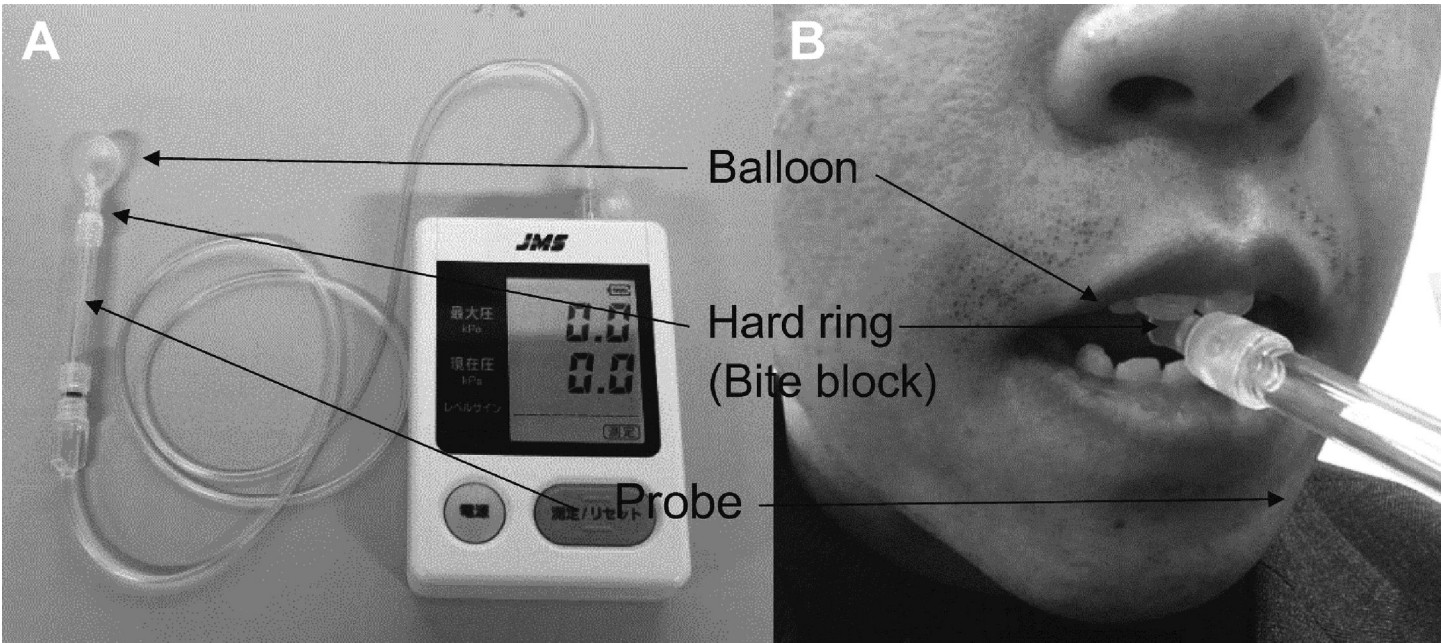

**Fig 1.** (A) Balloon-type tongue pressure measurement device (TPM-01; JMS Co. Ltd, Hiroshima, Japan), and (B) intra-oral positioning of the balloon.

measuring device, a balloon, which is fixed in front of the tongue, is compressed with the tongue toward the palate. Thus, it is speculated to reflect the anterior tongue strength. In addition, since the bite block is fixed by the incisors, the compressive force of the temporomandibular joint does not affect the MTP measurement.

## Speech analyses

The speech test was evaluated using two tasks: reading aloud a long sentence "The north wind and the sun" and oral-DDK of the lingual–alveolar consonant /ta/ (/a/ corresponds to /ʌ/ in Japanese). In the reading aloud task, patients were instructed to "read aloud using the volume, pitch, and speed as when normally speaking (without intentionally speeding up or slowing down)." A practice reading was performed before recording. In oral-DDK, the participants were instructed to "repeat /ta/ at maximum speed without taking a breath," and two measurements were performed. The speech of the participants was saved as an uncompressed file, with a sampling frequency of 44.1 kHz with 16-bit quantization using a digital voice recorder (R-05; Roland, Shizuoka, Japan). The recording was conducted in a quiet room with a noise level of 30 dBA or less. The microphone to mouth distance was 15 cm, and the input level was kept constant.

**Auditory perceptual assessment.** We evaluated speech intelligibility, which is one of the most important indicators of speech disorder severity. Three certified SLHTs (TT, YT, and YW) blindly and audibly evaluated the recording of long sentences that were read aloud. The representative value of speech intelligibility was the average value from the three evaluators on a nine-point speech intelligibility scale. This evaluation system is widely used in Japan, and its reliability was confirmed previously [30–32]. Speech intelligibility was scored from 1 to 5 in 0.5 increments. A score of 1 indicates normal; 5, severe; and 2–4, cases with speech disorders between the two points.

**Oral-DDK rate.** To reduce the effects of speech irregularities such as freezing, slurring, or syllable prolongation during speech onset, or respiratory dysfunction during the second half of

the task, we extracted ~3 s of recorded data from the middle parts of the audio for the analysis. The acoustic analysis was performed by the author TT using the acoustic analysis software Multi-Speech 3700 with Motor Speech Profile 5141 (KayPENTAX, Lincoln Park, NJ, USA). The maximum repetition rate of oral-DDK (unit: syllables/s) was calculated. The representative value of the oral-DDK is the average value of the two measurements.

**Second formant transition.** The acoustic analysis was performed by the author TT using Praat acoustic analysis software (ver. 6.0.50; Boersma & Weenink, University of Amsterdam). From the recorded voice of the long sentence "The north wind and the sun," the following three parts were analyzed: /ai/ and /jo/ of "太陽" /taijo:/ means "sun," and /ai/ of "外套"/ɡaito:/ means "coat." /taijo:/ and /ɡaito:/ appeared three times each in the long sentence, which were all analyzed (3 parts × 3 times = 9 times in total). For the analysis object, the transition section of the F2 of the two-vowel and semivowel sequences was extracted, and the F2 movement part was used as the measurement target. Details of the measurement targets: 1) /taijo/'s /ai/ transition, 2) /taijo/'s /jo/ glide, and 3) /ɡaito:/'s /ai/ transition. Fig 2 shows an example of the /taijo:/ measurement. F2 movement was measured based on a previous report measuring diphthong /aɪ/ [33] and semivowel /jæ/ [22]. The movement duration (ms) was set from the start (F2 onset) to the end (F2 offset) of the F2 movement. The F2 linear predictive coding tracks on a wide-band spectrogram (analysis bandwidth, 300 Hz) were manually edited and identified. The 20/20 rule (specifying a frequency change of ≥20 Hz during 20 ms in the transition onset and offset) was applied [34]. If the F2 track was unclear due to hoarseness or other noises, we identified it by referring to the changes in F1 (opening and closing of the mandible) that were almost synchronized with F2 in the /ai/ and /jo/ sequences. Thus, relatively stationary portions of the vowel before or after the target transition were not included in the analysis. Movement extent (Hz) is the difference in the F2 between the beginning and end. The F2 slope (unit: Hz/ms) was obtained by dividing the movement extent by the movement duration. Therefore, the F2 slopes are expressed in absolute values (Hz/ms) throughout this manuscript to eliminate the positive/negative sign caused by the target-inherent F2-movement direction. Based on a previous study of dysarthria [19], we averaged all the F2 slopes with different acoustic properties. This aimed to reduce the error from the actual syllable-specific tongue motion velocity, considering the possibility that mandibular opening and closing could be related to the context [22].

## Statistical analyses

Data are expressed as the median (minimum, 25th percentile, 75th percentile, maximum). SPSS v. 27.0 (IBM Corp., Armonk, NY, USA) was used for the statistical analyses. Nonparametric tests were selected for smaller sample sizes in the control group and for effect size comparisons between indicators. The Mann–Whitney U test was used to assess the difference between a speaker with dysarthria and a neurologically normal speaker. The effect size of the difference was evaluated by calculating r from the obtained z-value and the number of participants. The χ-square test was used to evaluate the differences in the sex ratio.

Pearson correlation was used to examine the relationships between continuous variables and MTP results for speakers with dysarthria. Spearman rank correlation analysis indicated the strength of the associations between the MTP and speech intelligibility because of non-normal distributions for speakers with dysarthria. Normal distribution was confirmed using the Shapiro–Wilk test. Since a difference exists between male and female individuals in the MTP [28] and F2 [35], subgroup analysis was also performed. In addition, a subgroup correlation analysis by the dysarthria subtype and MTP severity (divided into two groups by median) was performed. For subgroup analysis, a nonparametric method was selected from a small sample

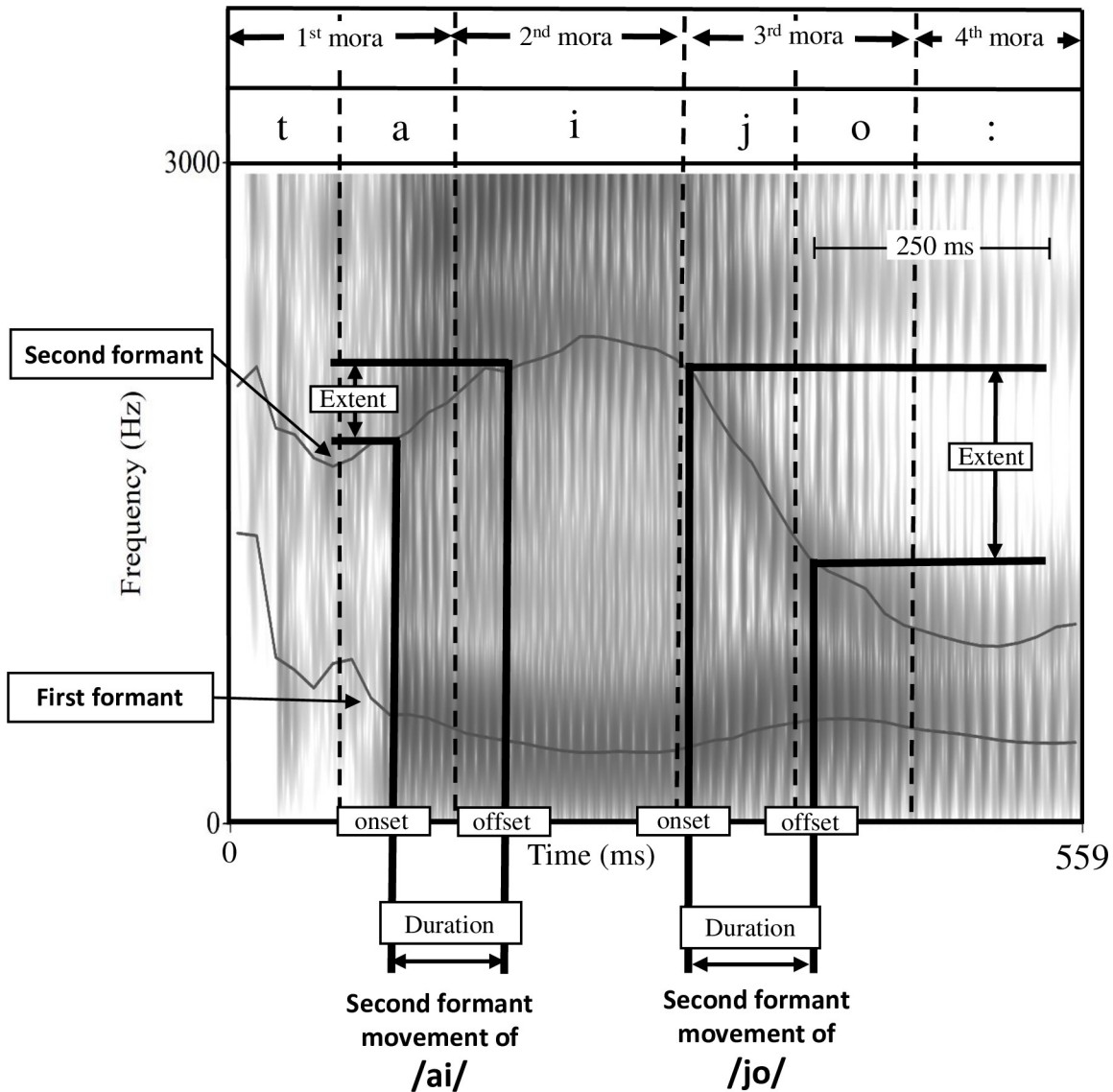

**Fig 2. A spectrogram of the word /taijo:/ produced by a man with stroke and mild unilateral upper motor neuron dysarthria.** This figure illustrates a major two vowel sequence and semivowel F2 movements in /ai/ and /jo/. The black lines running through the estimated centers of the first and second formants (F1 and F2) illustrate the formant tracing. Duration is the time interval of the F2 movement. Extent is the range of frequency change in the F2 movement. The F2 movement for each syllable is determined by a change in frequency of at least 20 Hz during a 20 ms period, not including the relatively stationary portions before and after the syllable.

size. Statistical significance was set at $p < 0.05$. The sample size required to detect an association between speech-related variables and the MTP was 59 or more participants, with an effect size of 0.35 and a power of 0.8, based on a previously established methodology [13]. In a previous study [13], the effect size of anterior tongue pressure and auditory perceptual assessment ranged from 0.35 to 0.52.

Cronbach's α was used to evaluate the reliability of the evaluators of speech intelligibility. The in-session reproducibility of the oral-DDK rate within the participants was evaluated using intraclass correlation coefficients (ICCs). The ICC was used to evaluate the reliability within measurement of the F2 slope. It evaluated a group of speakers with dysarthria who may have more variability in measurements than healthy individuals.

## Statement of ethics

Consent from participants was obtained in writing and verbally. Informed consent was obtained from all participants. All procedures were approved by the ethics committees of the Uonuma Kikan Hospital (Approval no.: 30–007) and Nagaoka Nishi Hospital (Approval no.: 29–02). Consent was also obtained from all participants regarding the secondary use of data. We guaranteed the participants of their rights to withdraw from the study using an opt-out procedure.

## Results

The final analysis included 63 participants (median age, 68 years; interquartile range, 58–77 years; 44 men and 19 women) (Table 1). There were 72 entries in this study. However, seven speakers with severe cognitive impairments were excluded. Sixty-five speakers with dysarthria met the eligibility criteria. Of these, two were excluded due to lost data. A significant difference in the age and sex was found between neurologically normal speakers and speakers with dysarthria (age: $p < 0.001$, sex: $p < 0.031$). Details of all the participants, including healthy speakers, are shown in the S1 File.

The diagnoses were as follows: neurovascular events (41), progressive neurological diseases (16) (amyotrophic lateral sclerosis, 3; multiple system atrophy, 5; Parkinson's disease, 2; myasthenia gravis, 2; corticobasal degeneration, 1; progressive supranuclear palsy, 1; spinocerebellar ataxia, 1; and hereditary spastic paraplegia, 1), and other neurological diseases (6). The breakdown of the dysarthria subtypes in the cases was as follows: spastic, 3; flaccid, 7; ataxic, 7; hypokinetic, 6; hyperkinetic, 1; unilateral upper motor neuron, 16; mixed, 14; and undetermined, 9.

**Table 1. Demographic information of speakers with dysarthria (n = 63).**

| Factors | |
|---|---|
| **Sex (male/female)** | 44/19 |
| **Age (years)** | 68 (24, 58, 77, 86) |
| **Height (m)** | 1.62 (1.43, 1.55, 1.68, 1.81) |
| **Weight (kg)** | 61.0 (32.9, 50.8, 69.1, 112.1) |
| **Body mass index (kg/m$^2$)** | 22.5 (13.9, 19.7, 25.7, 46.1) |
| **Albumin (g/dL)** | 3.8 (2.9, 3.5, 4.2, 4.9) |
| **Etiologies (%)** | |
| Neurovascular event | 41 (65.1) |
| Progressive neurological disease | 16 (25.4) |
| Other neuropathies | 6 (9.5) |
| **Dysarthria subtype (%)** | |
| Spastic | 3 (4.8) |
| Flaccid | 7 (11.1) |
| Hypokinetic | 6 (9.5) |
| Hyperkinetic | 1 (1.6) |
| Ataxic | 7 (11.1) |
| UUMN | 16 (25.4) |
| Mixed | 14 (22.2) |
| Undetermined | 9 (14.3) |

Data are expressed as medians (minimum, 25th percentile, 75th percentile, maximum) for continuous variables (age, height, weight, body mass index, and albumin) and as frequencies (percentages) for categorical variables (etiologies and histories of the dysarthria subtype). UUMN, unilateral upper motor neuron.

**Table 2. Maximum tongue pressure and speech measures of speakers with dysarthria (n = 63) and neurologically normal speakers (n = 30).**

| Parameters | Symbols | Speakers with dysarthria | Younger neurologically normal speakers | Effect size (r) |
|---|---|---|---|---|
| **MTP** | kPa | 33.3 (9.3, 24.9, 39.1, 54.7)** | 44.2 (13.7, 32.7, 48.7, 73.3) | -0.38 |
| **Speech intelligibility** | - | 2.0 (1.0, 1.5, 2.2, 5.0)** | 1.0 (1.0, 1.0, 1.0, 2.0) | -0.76 |
| **/ta/ DDK rate** | syllable/s | 5.3 (2.4, 4.1, 6.2, 9.2)** | 7.6 (5.2, 7.0, 8.6, 9.6) | -0.69 |
| **F2 slope** | Hz/ms | 8.8 (3.6, 6.6, 11.1, 18.3)** | 11.0 (5.2, 9.9, 12.7, 22.3) | -0.41 |

Data are expressed as medians (minimum, 25th percentile, 75th percentile, maximum).

**p < 0.001. DDK, diadochokinesis; F2, second formant; KPa, kilopascal; MTP, maximum tongue pressure. /ta/ is a syllable pattern comprising a lingual–alveolar consonant with a vowel.

## Tongue strength and speech measures

Table 2 shows the results of the MTP and speech-related evaluations measured in this study.

The interrater reliability assessed using Cronbach's α for speech intelligibility among the three raters was 0.931, indicating high reliability. The two measurements of the oral-DDK rate showed high reproducibility of the ICC (0.985). In addition, of the 63 speakers with dysarthria, the recorded speeches of 19 randomly selected speakers (30.2%) were measured again by the same examiner who had performed the first measurement (6 months previously), and the ICC of the two measurements showed high reliability (F2 slope, 0.904).

## Relationship between tongue strength and speech measures

The MTP and F2 slope were significantly associated among speakers with dysarthria (r = 0.368, p = 0.003). No significant correlation was detected between the MTP and oral-DDK/speech intelligibility (Table 3).

In the analysis by sex groups, a significant correlation was observed between the MTP and F2 slope (male, rs = 0.397, p = 0.008; female, rs = 0. 479, p = 0.038) (Fig 3). The correlation between MTP and speech intelligibility was significant only in males (rs = -0. 328, p = 0.030). There was no significant correlation between MTP and the /ta/ DDK rate in either sex.

In the analysis by the dysarthria subtype, a significant correlation was observed between the MTP and F2 slope (Flaccid, rs = 0.786, p = 0.036; Mixed, rs = 0.640, p = 0.014; Table 4). There was no significant correlation between any of the combinations in the other subtypes. Note that spastic (n = 3) and hyperkinetic (n = 1) types were excluded from the analysis due to their small sample size.

In the analysis by the groups categorized according to maximum tongue pressure, a significant correlation was observed between the MTP and all speech-related variables only in the

**Table 3. Correlation coefficients (two sided) between maximum tongue pressure and speech measures for speakers with dysarthria (n = 63).**

| Speech measures | Correlation coefficients | p value |
|---|---|---|
| **Speech intelligibility** | rs = -0.191 | 0.134 |
| **/ta/ DDK rate** | r = 0.142 | 0.266 |
| **F2 slope** | **r = 0.368*** | 0.003 |

Value in bold and with asterisk indicates that r is significant (p < 0.01) (two-tailed). Correlation analysis used Spearman's rank correlation coefficient for intelligibility and Pearson's correlation coefficient for other indicators. /ta/ is a syllable pattern comprising a lingual–alveolar consonant with a vowel. DDK, diadochokinesis; F2, second formant.

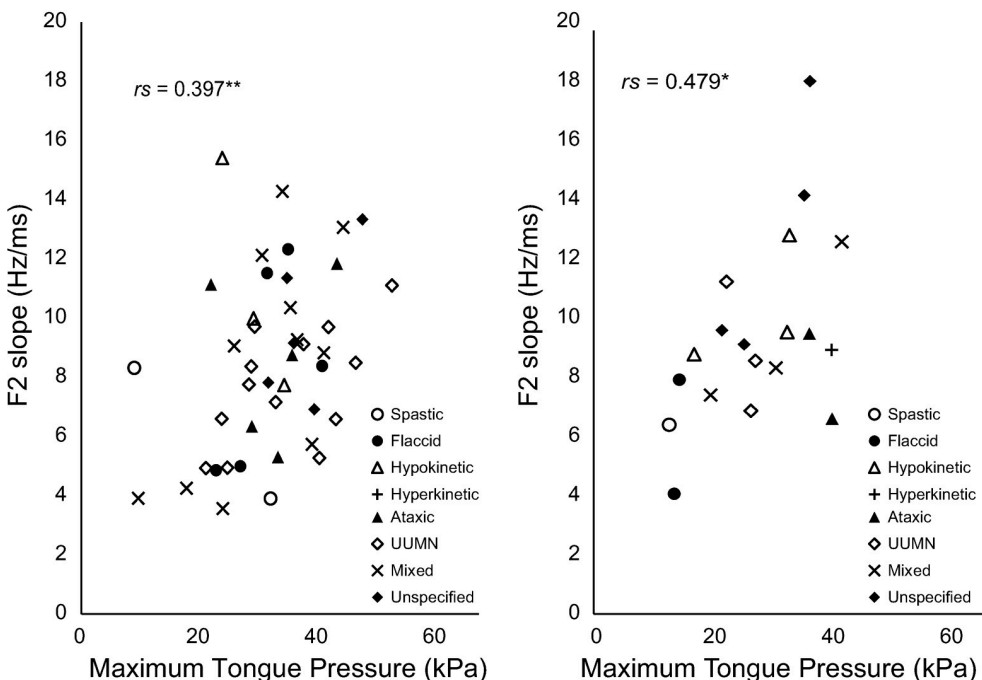

**Fig 3.** Bivariate scatter plot of maximum tongue pressure and second formant (F2) slope in male speakers with dysarthria (left) and female speakers with dysarthria (right). UUMN, unilateral upper motor neuron. *p < 0.05, **p < 0.01. The plot shows the subtypes of each speaker with dysarthria. White circles, spastic; black circles, flaccid; white triangles, hypokinetic; crosses, hyperkinetic; black triangles, ataxic; white diamonds, UUMN; X, mixed; and black diamonds, unspecified types.

lower MTP group (Speech intelligibility, rs = 0.397, p = 0.008; /ta/ DDK rate, rs = 0. 479, p = 0.038; F2 slope, rs = 0. 479, p = 0.038; Fig 4).

## Discussion

In this study, we measured the tongue pressure in speakers with various types of dysarthria, following which, we conducted a correlation analysis between MTP and speech-related

**Table 4. Spearman's rank correlation coefficients (two sided) between the maximum tongue pressure and speech measures for each subtype of speakers with dysarthria (Total n = 59).**

| Dysarthria subtype groups | Speech intelligibility | /ta/ DDK rate | F2 slope |
|---|---|---|---|
| **Flaccid (n = 7)** | rs = -0.436<br>p = 0.328 | rs = 0.517<br>p = 0.180 | **rs = 0.786***<br>p = 0.036 |
| **Hypokinetic (n = 6)** | rs = -0.232<br>p = 0.658 | rs = -0.314<br>p = 0.544 | rs = -0.257<br>p = 0.623 |
| **Ataxic (n = 7)** | rs = 0.018<br>p = 0.969 | rs = 0.429<br>p = 0.337 | rs = 0.321<br>p = 0.482 |
| **UUMN (n = 16)** | rs = -0.050<br>p = 0.855 | rs = -0.218<br>p = 0.418 | rs = 0.321<br>p = 0.225 |
| **Mixed (n = 14)** | rs = -0.146<br>p = 0.618 | rs = 0.249<br>p = 0.391 | **rs = 0.640***<br>p = 0.014 |
| **Undetermined (n = 9)** | rs = 0.052<br>p = 0.894 | rs = -0.033<br>p = 0.932 | rs = 0.201<br>p = 0.604 |

Values in bold and with asterisks indicate that rs is significant (p < 0.05) (two-tailed). /ta/ is a syllable pattern comprising a lingual–alveolar consonant with a vowel. DDK, diadochokinesis; F2, second formant; UUMN, unilateral upper motor neuron.

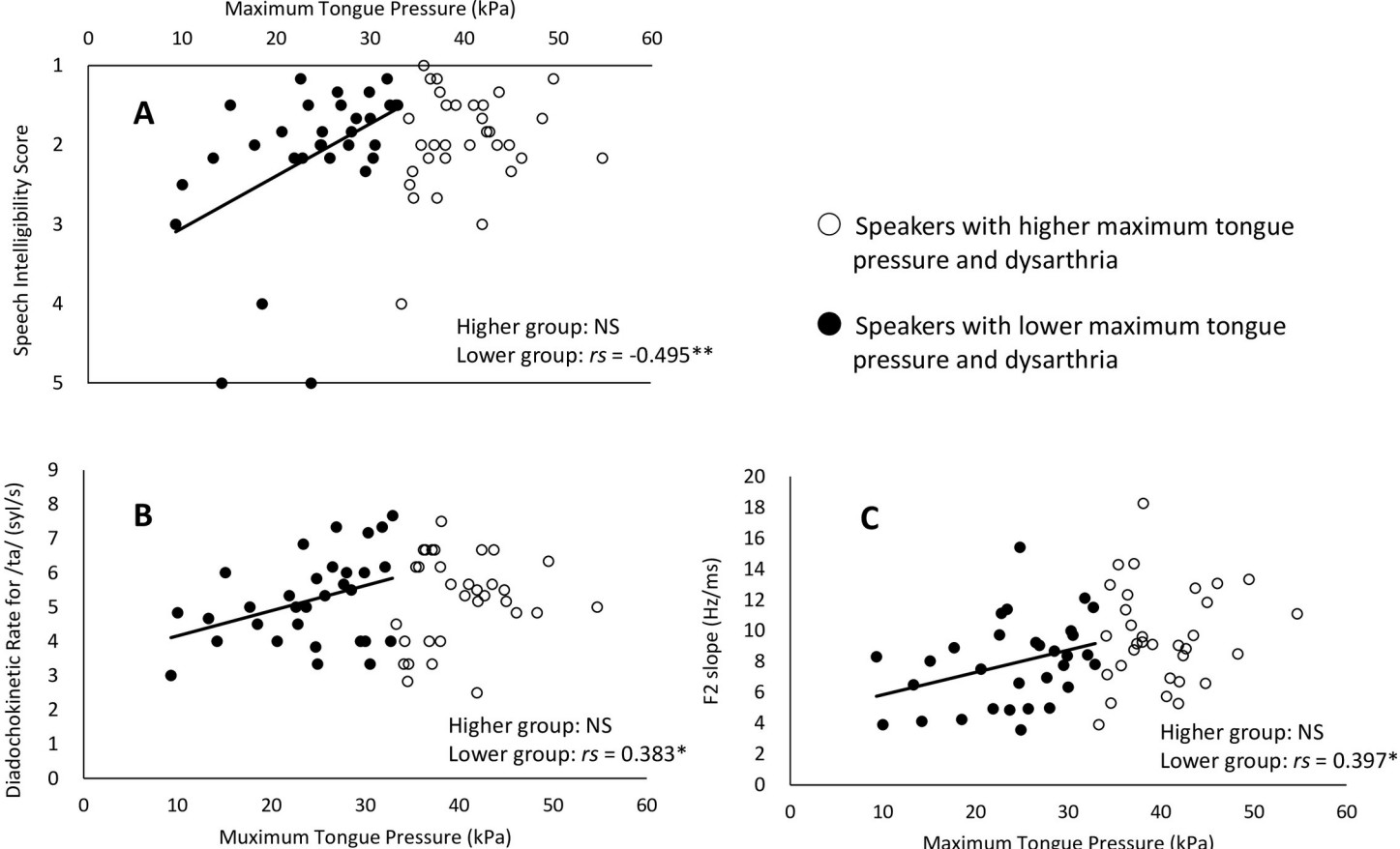

**Fig 4. Bivariate scatter plot and best-fit regression line of the maximum tongue pressure and speech-related variables in the two speakers with dysarthria groups divided according to the maximum tongue pressure by median.** (A) speech intelligibility score, (B) diadochokinetic rate for /ta/, (C) second formant (F2) slope. $^*p < 0.05$, $^{**}p < 0.01$. Speakers with lower maximum tongue pressure and dysarthria (lower group) tended to have lower speech-related variables, such as lower maximum tongue pressure.

indices. The tongue pressure was not significantly associated with the oral-DDK rate and speech intelligibility. However, the tongue pressure was significantly associated with the F2 slope. In addition, a significant difference in the MTP and all speech-related indicators was noted between speakers with dysarthria and neurologically normal speakers, with moderate to large effects. To date, many studies have investigated the relationship between tongue muscle strength, speech intelligibility [36], articulatory precision [4,13,37], and the oral-DDK rate [11,24,38]. To the best of our knowledge, this study is the first to investigate the relationship between tongue strength and the F2 slope. These findings provide some implications for understanding tongue muscle strength in patients with dysarthria.

The effect size of the difference in the oral-DDK rate between speakers with dysarthria and those in the control group was large. However, in speakers with dysarthria, the oral-DDK rate and MTP were not correlated. Previous studies have shown contradictory results regarding the relationship between the oral-DDK rate and tongue strength. For example, in patients with oculopharyngeal muscular dystrophy (n = 12), the MTP and oral-DDK rates were significantly reduced compared with healthy participants, although no correlation was found [11]. On the other hand, conventional speech rehabilitation therapy and tongue strength exercises for speakers with cerebrovascular disorders result in a significantly faster oral-DDK rate at /tʌ/ [24]. One of the factors behind these contradictory results is the inadequacy and imbalance of

the participants [10,13]. Specifically, depending on the threshold of the tongue muscle weakness affecting speech, the correlation may not be clear when only a small number of patients has the most severe muscle weakness [11,13]. In addition, there may be a difference in the degree of contribution to the speech impairment between diseases in which muscle weakness is the main symptom [4,10] and other diseases [3,13]. A previous study including speakers with different types of dysarthria (n = 55) described a weak correlation between the oral-DDK rate for /tʌ/ and MTP (r = 0.247) [13]. In the current study, not much difference was observed in the correlation coefficient between the MTP and oral-DDK rate (r = 0.142) compared with a previous similar study. Therefore, the results of this study do not support a strong relationship between MTP and the oral-DDK rate.

In this study, the MTP was not correlated with speech intelligibility in speakers with dysarthria. A previous study described a moderate correlation between speech intelligibility (%) and the MTP (rs = 0.349) in speakers with different types of dysarthria (n = 55) [13]. It also showed that the subgroups (n = 8) with severely reduced MTP included three participants with amyotrophic lateral sclerosis and five with sustained combat injuries (many with polytraumatic injuries). This may have affected the results, as our study did not include participants with sustained combat injuries. Combat injuries with orofacial injuries may have a greater impact on the tongue function necessary for speech. In addition, speech intelligibility is evaluated using auditory (qualitative) techniques and is not highly sensitive to mild cases [38]. Therefore, the difference in the correlation coefficient could not possibly detect a significant correlation because few participants with severe dysarthria were included in this study. Furthermore, comparing our study with previous studies [4,10], which showed a strong correlation between word intelligibility and tongue muscle strength, the heterogeneity of the target disease is considered to have an effect. For example, Searl et al. found that 13 participants with amyotrophic lateral sclerosis (n = bulbar type 8, spinal type 5) have a strong correlation between tongue strength and speech intelligibility [10]. Amyotrophic lateral sclerosis is correlated with mixed flaccid–spastic type of dysarthria, and muscle weakness has a significant effect on speech [1]. Therefore, in cases where muscle weakness extends to the whole body, the relationship between speech intelligibility and tongue muscle strength becomes stronger. In our results, speech intelligibility and oral-DDK rate were not significantly correlated with MTP in all subtypes. However, the flaccid type (n = 7) had the highest correlation coefficient among all types (speech intelligibility, rs = -0.436; /ta/ DDK rate, rs = 0.517). Additional subtype-specific studies are warranted.

In this study, the F2 slope and MTP were significantly correlated in speakers with dysarthria (r = 0.368). Solomon et al. concluded that auditory articulatory precision is suitable for assessing the association between tongue strength and speech function [13]. The main cause of reduced speech intelligibility is abnormal articulation [14], although speech intelligibility and articulatory precision are not always the same [39]. Focusing on the adequacy of articulation is reasonable to more directly evaluate the tongue, which plays a major role in articulation. F2 roughly corresponds to the back-and-forth movement of the tongue [15]. The F2 slope, calculated from the movement duration and extent of the F2, is an acoustic index that correlates with the perceived accuracy of vowels [16]. The F2 slope is speculated to reflect the speed of the tongue movement during articulation. In this study, the correlation between the F2 slope and tongue muscle strength was similar to all speakers with dysarthria in the sex-separated subgroups. In previous studies, sex does not significantly affect the relationship between the F2 slope and speech intelligibility in speakers with dysarthria [22,40]. As for the MTP, sex differences disappear beyond the age of 50 years [28]. In summary, in speakers with dysarthria, tongue strength may be associated with tongue movement velocity during articulation. Our

findings partially support the hypothesis that "muscle weakness is associated with slow speech" [2].

On the other hand, the effect size of the difference between the speakers with dysarthria and healthy participants on the MTP and F2 slope in this study was moderate. However, the effect of the difference in oral-DDK rate and speech intelligibility between healthy speakers and speakers with dysarthria was large. In a previous study, Japanese speakers with mild dysarthria (n = 16) showed no significant difference in the MTP compared with speakers without dysarthria (n = 29) [41]. In addition, the F2 slope was lower in speakers with moderate to severe dysarthria [21,34]. Thus, MTP and F2 slope are less capable of differentiating between healthy speakers and speakers with mild dysarthria than oral-DDK rate and speech intelligibility. Nevertheless, the findings of this study showed a moderate correlation between the F2 slope and MTP in speakers with dysarthria. The F2 slope is correlated with tongue muscle strength, which is stronger than the relationship between tongue muscle strength and other speech-related indicators.

In addition, the present study showed different results for the F2 slope and oral-DDK rate, which are related to tongue movement speed during articulation. This result is worthy of special mention. The DDK rate of /ta/ also depends on the speed of the voice on/off and mandibular opening/closing. In addition, the oral-DDK rate counts the number of syllables produced per second. Some articulation distortions (i.e., under shooting) are not reflected in the DDK rate measurements. Thus, both clear and unclear articulations are counted as one syllable (i.e., narrow or wide range of movement). The F2 slope, on the other hand, acoustically isolates the back-and-forth movement of the tongue during articulation and measures its speed. In a previous study, an oral-DDK task was not possible at maximum articulatory velocity [42]. Therefore, it is likely that the F2 slope better reflects the speed of the tongue movements than the oral-DDK rate, and this may have affected the results.

This study has some limitations. First, it was a cross-sectional study; thus, the causal relationship between tongue muscle strength and the F2 slope is unknown. Longitudinal studies are useful for verifying causal relationships. Second, it did not consider the speech treatments the participants received in this study (e.g., non-speech oral motor exercises, speech rate modification, and Lee Silverman voice treatment). Third, the control group was not strictly set. In this exploratory research, we included young people to clearly show the difference between speakers with dysarthria and healthy speakers. A comparative controlled study involving participants matched for age and sex would be useful for a more detailed understanding. Finally, because only few participants with severe dysarthria were included, the data were not sufficient to adapt to people with severe disease. JMS devices have bite blocks and cannot be used in patients with missing teeth, which indicates that collecting data from elderly and severely ill patients may have been difficult. For studies involving speakers with dysarthria that require those with severe dysarthria, considering devices without bite blocks and sensor-type pressure gauges is necessary. Hence, further studies are warranted to overcome these limitations and to assess the broader applicability of the MTP and F2 slope.

## Conclusions

The MTP was not significantly correlated with the oral-DDK rate and speech intelligibility. However, using correlation analysis, we confirmed that the F2 slope and MTP were related. This suggests that the maximum isometric tongue strength is associated with the tongue movement speed during articulation. From a clinical point of view, tongue strength training may be considered for dysarthria speakers with a reduced F2 slope (that is, the appropriateness of articulation and speed) and tongue strength. From a research point of view, the F2 slope may

be useful for verifying the effect of tongue strength training and the appropriateness of articulation and speed.

## Supporting information

**S1 File. Dataset with all the participants.** Re-measurement: Result of a re-measurement by the same inspector > 6 months after the original measurement (30% of speakers with dysarthria).
(XLSX)

## Acknowledgments

The authors would like to thank Nagaoka Nishi Hospital (Nanae Koyama, Sanae Saito, and Hiromi Takeuchi) and Uonuma Kikan Hospital (Yuuki Kobayashi and Koki Maruyama) for their assistance with data collection.

## Author Contributions

**Conceptualization:** Toshiaki Tamura, Yasuhiro Tanaka, Yoshihiro Watanabe, Katsuro Sato.

**Data curation:** Toshiaki Tamura, Yasuhiro Tanaka, Yoshihiro Watanabe.

**Formal analysis:** Toshiaki Tamura.

**Funding acquisition:** Toshiaki Tamura.

**Investigation:** Toshiaki Tamura, Yoshihiro Watanabe.

**Methodology:** Toshiaki Tamura, Yasuhiro Tanaka, Katsuro Sato.

**Project administration:** Toshiaki Tamura, Yasuhiro Tanaka, Yoshihiro Watanabe, Katsuro Sato.

**Resources:** Toshiaki Tamura, Katsuro Sato.

**Supervision:** Yasuhiro Tanaka, Katsuro Sato.

**Validation:** Toshiaki Tamura, Yasuhiro Tanaka, Yoshihiro Watanabe, Katsuro Sato.

**Visualization:** Toshiaki Tamura, Yasuhiro Tanaka.

**Writing – original draft:** Toshiaki Tamura, Yasuhiro Tanaka.

**Writing – review & editing:** Toshiaki Tamura, Yasuhiro Tanaka, Yoshihiro Watanabe, Katsuro Sato.

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
