## [Decision Letter · Decision Letter 0]

19 Nov 2021

PONE-D-21-32058Relationships between maximum tongue pressure and second formant transition in speakers with different types of dysarthriaPLOS ONE

Dear Dr. Tamura,

Thank you for submitting your manuscript to PLOS ONE. After careful consideration, we feel that it has merit but does not fully meet PLOS ONE’s publication criteria as it currently stands. Therefore, we invite you to submit a revised version of the manuscript that addresses the points raised during the review process.

Please see the editor and reviewer comments, below.

We look forward to receiving your revised manuscript.

Kind regards,

Sara Finley, Ph.D.

Academic Editor

PLOS ONE

Journal Requirements:

“This work was supported by the Japan Society for the Promotion of Science Grants-in-Aid for Scientific Research (KAKENHI) (https://www.jsps.go.jp/j-grantsinaid/) under Grant number JP20K19324.”

Additional Editor Comments:

I have received 2 reviewer reports for the paper, “Relationships between maximum tongue pressure and second formant transition in speakers with different types of dysarthria”. Both reviewers note that the research question is of potential interest to a wide range of scholars, and that the methods and interpretation of results are generally sound. However, both reviewers expressed some concerns related to the details of how the F2 slope was calculated. Given that this is the major contribution of this paper, I agree that more details need to be included in your revisions about how F2 slope was calculated, particularly the rationale for the method chosen. For example, Reviewer 2 asks questions about whether the F2 slope was calculated across the entire word, or within each vowel separately, so more details on how the vowels in the target words were analyzed distinctly from the consonants in the words is needed. An additional figure, as suggested by reviewer 1, would be welcome. Reviewer 1 also suggests several places where the writing could be improved for clarity, and I suggest following this advice, particularly where redundancies can be avoided (also noted by Reviewer 2), or jargon can be removed or explained (particularly in the abstract). Reviewer 2 also suggests that some analysis be conducted related to the type and severity level of dysarthria, which would be welcome, given the title of the paper.

Reviewers' comments:

Reviewer's Responses to Questions

**Comments to the Author**

1. Is the manuscript technically sound, and do the data support the conclusions?

Reviewer #1: Yes

Reviewer #2: Partly

2. Has the statistical analysis been performed appropriately and rigorously? 

Reviewer #1: Yes

Reviewer #2: Yes

3. Have the authors made all data underlying the findings in their manuscript fully available?

Reviewer #1: Yes

Reviewer #2: Yes

4. Is the manuscript presented in an intelligible fashion and written in standard English?

Reviewer #1: Yes

Reviewer #2: Yes

5. Review Comments to the Author

Reviewer #1: Relationships between maximum tongue pressure and second formant transition in　speakers with different types of dysarthria

General comments:

This paper has a new important issue on speech evaluation for dysarthria patients. But the outside reader of your manuscript it could be a little hard to understand correctly. Could you look at the points below:

Abstract

1. P3 L22-23

“The tongue pressure in speakers with dysarthria was significantly associated with the second formant transition slope.”

This sentence seems to have the same meaning as the one before this sentence. I recommend that you delete this sentence.

2. P3 L23-24

“This result suggests that the maximum isometric tongue strength is associated with articulation severity and tongue movement speed during articulation.”

The results showed that MTP was not correlated with speech intelligibility. The "the maximum isometric tongue strength is associated with articulation severity" in this sentence is inconsistent with the result.

Introduction

1. P5. L56-57

“these are qualitative assessments”

Does this “these” mean "Elevation strength of the anterior tongue"? If so, the elevation strength of the anterior tongue is quantitative assessment, not qualitative assessment.

2. P3 L57

“they affect the distribution of data (ceiling or floor effect).”

Please add references.

3. P6 L67-69

“The hypothesis that … oral-DDK rates.”

Please add references.

Material and Methods

1. P9 L121

“thickness, 0.5mm”

I think it would be more appropriate to write the diameter rather than the thickness of the bite block.

2. P12 L177-179

I suggest you should make a figure for analysis methods about onset time, offset time, movement duration, and movement extent.

3. P12 L176-177

In the previous study, /aɪ/ was used to analyze the F2 movement. Why did you include /jo/ as well as /ai/ in the analysis in this study?

4. P12 L184-185

The F2 slope values of /ai/ and /jo/ were calculated as one average value in this study. Why did you calculate the two types of two-vowels (/ai/ and /jo/) with different acoustic characteristics together as one average value?

Results

1. P14 L223-224

This cross-sectional study population included speakers with dysarthria who were

admitted to acute and convalescent hospitals between September 2017 and June 2020. Please describe how many participants were entered, and how many participants did not match the inclusion criteria and were excluded, resulting in 65 participants matching the inclusion criteria. In addition, please describe the reasons for the exclusion.

2. P14-15 L225-227

“A significant difference in age was found between neurologically normal speakers and speakers with dysarthria (p < 0.001)”.

This sentence only showed the difference in age between neurologically normal speakers and speakers with dysarthria. Please describe the statistically analyzed differences in sex between neurologically normal speakers and speakers with dysarthria.

3. P17 L244

“The average MTP of a speaker with dysarthria was 32.3 ± 9.9 kPa.”

The median of MTP was shown in table 2, so this statement did not need to show the mean of MTP.

Discussion

1. P21 L303-304

“One of the factors behind these contradictory results is the inadequacy and imbalance of the subjects [10,13].”

Please add a specific explanation for “inadequacy and imbalance of the subject”.

2. P21 L309-310

“Therefore, the results of this study support the weak correlation between the oral-DDK rate and tongue muscle strength.”

In this study, there was no significant correlation between the oral-DDK rate and tongue muscle strength. I consider that this statement is over-interpreted.

3. P21 L316-317

“This may have affected the results, as our study did not include participants with sustained combat injuries.”

Please describe how the non-inclusion of the participants with sustained combat injurie affected the results in this study.

4. P22 L321

were included →　were included in this study.

5. P23 L354-355

“Therefore, the smaller effect size of the difference between the F2 slope and MTP suggests that speakers with mild dysarthria have lower discriminative function.”

This sentence is difficult to understand. Please describe why the smaller effect size of the difference between the F2 slope and MTP suggests that speakers with mild dysarthria have lower discriminative function.

6. P24 L357

between the F2 slope and MTP. →　between the F2 slope and MTP in speaker with dysarthria.

7.

I consider that the oral-DDK rate and F2 slope are associated with the tongue movement speed during articulation. However, the MTP was significantly related to the F2 slope but not significantly related to the oral-DDK rate in this study. Please describe why the oral-DDK rate and F2 slope, which are associated with the tongue movement speed during articulation, had different results from each other.

Reviewer #2: This paper investigates the relationships between maximum tongue pressure and speech-related features, which include speech intelligibility, /ta/ DDK, and F2-slope. While speech intelligibility and /ta/ DDK were revealed to have no significant correlations with maximum tongue pressure, F2-slope was significantly correlated with the maximum tongue pressure (r=0.37, p<0.05). This paper proposes that the F2- slope may be useful at verifying the effect of tongue strength training.

1. Is the manuscript technically sound, and do the data support the conclusions?

- The idea of using F2-slope which can see both articulation accuracy and articulation rate, is convincing.

- F2-slope is usually used for analyzing diphthongs(one vowel), and sometimes vowel sequences. However, I am concerned about this experiment design, which analyze the F2-slope in a word level. Are the features extracted from the start of /a/ and end of /o/? If this is the case, the design should be revised in a major manner. Especially, for word /gaito/, the consonant is in between the two vowels, which must interrupt in extracting the F2-slope. Even for /taijo/, the difference of F2 between /a/ and /i/ are much larger compared to difference of F2 between /a/ and /o/. Analyzing the F2-slope for each vowel sequence/semi-vowel may be more persuasive.

2. Has the statistical analysis been performed appropriately and rigorously?

Mann-Whitney U test is applied to investigate the difference between healthy speakers and dysarthric speakers. Pearson Correlation is used to examine the relationships between MTP results and speech function features. Subgroup analysis by gender are appropriately performed.

Further subgroup analysis by dysarthric subtypes and severity levels should also be considered. In particular, the authors argue that the reason the study results do not agree with the previous results is because of the different distribution of speakers. Hence this analysis is necessary.

3. Have the authors made all data underlying the findings in their manuscript fully available?

- The datasheet used for the statistical analysis is uploaded. All features used in the analysis are included - TP, F2-slope, /ta/DDK, speech intelligibility.

- However, each notation must be explained of its meaning. For example, what does POST and typeNo imply?

- Further, each notations should be explained why each feature is included in the data. It is hard to understand why are height, weight, BMI, Albumin are included for the analysis.

- Brief information of dysarthric speakers should be stated in the manuscript. (subtypes)

4. Is the manuscript presented in an intelligible fashion and written in standard English?

- The manuscript is presented in an intelligible fashion and well-written in standard English.

- Though there are some redundant sentences that should be deleted for the final submissions.

- Abstract is quite confusing, especially the last sentence : "However, based on the degree of these correlations, the hypothesis that the relationship between the maximum force of the tongue and speech function is weak is also strengthened." This sentence blurs the main idea of this paper, which suggests the significant correlation between maximum force of the tongue and F2-slope.

- line 338 : F2-slope -> F2 (second formant)

- line 340 : It -> F2-Slope

6. PLOS authors have the option to publish the peer review history of their article (what does this mean?). If published, this will include your full peer review and any attached files.

Reviewer #1: No

Reviewer #2: No

---

## [Author Response · Author response to Decision Letter 0]

12 Dec 2021

Joerg Heber 

Editor-in-Chief

Sara Finley, Ph.D.

Academic Editor

PLOS ONE

Dear Editor: 

I wish to re-submit the manuscript, titled “Relationships between maximum tongue pressure and second formant transition in speakers with different types of dysarthria.” The manuscript ID is PONE-D-21-32058.

I thank you and the reviewers for your thoughtful suggestions and insights. The manuscript has significantly benefited from these insightful suggestions. I look forward to working with you and the reviewers to move this manuscript closer to publication in PLOS ONE.

The manuscript has been rechecked and the necessary changes have been made in accordance with the reviewers’ suggestions and marked in red. The responses to all the comments have been prepared and presented below. 

Thank you for your consideration. I look forward to hearing from you.

Sincerely,

Toshiaki Tamura 

Department of Speech, Language, and Hearing Sciences, Niigata University of Health and Welfare, Niigata city, Niigata 950-3198, Japan

Tel: +81 (25) 257-4507

Fax: +81 (25) 257-4507

Email: toshiaki-tamura@nuhw.ac.jp

 

Point-by-point responses to comments from the Editor and Reviewers

Thank you for reviewing our work. We appreciate all your comments and suggestions. We have revised the manuscript accordingly. Our point-by-point responses are presented below.

Response to Journal Requirements

Comment 1: Please ensure that your manuscript meets PLOS ONE's style requirements, including those for file naming. The PLOS ONE style templates can be found at

Response 1: We are very grateful for your guidance. We have ensured that the manuscript meets the style requirements of PLOS ONE.

Comment 2: Thank you for stating the following financial disclosure:

“This work was supported by the Japan Society for the Promotion of Science Grants-in-Aid for Scientific Research (KAKENHI) (https://www.jsps.go.jp/j-grantsinaid/) under Grant number JP20K19324.”

Response 2: Thank you for your valuable comment. Kindly revise the data availability statement accordingly for us. We have also added a statement to the cover letter.

“This work was supported by the Japan Society for the Promotion of Science Grants-in-Aid for Scientific Research (KAKENHI) (https://www.jsps.go.jp/j-grantsinaid/) under Grant number JP20K19324. The funders had no role in the study design, data collection and analysis, decision to publish, and preparation of the manuscript.”

Comment 3: We note that you have stated that you will provide repository information for your data at acceptance. Should your manuscript be accepted for publication, we will hold it until you provide the relevant accession numbers or DOIs necessary to access your data. If you wish to make changes to your Data Availability statement, please describe these changes in your cover letter and we will update your Data Availability statement to reflect the information you provide.

Response 3: The data required for the analysis of this paper are included in the supplementary data. Therefore, we did not provide data repository information. Please change the data availability statement accordingly for us. We apologize for the inconvenience. We have also added a request for change in the cover letter.

“Furthermore, the data required for the analysis of this paper are included in the submitted supplementary data. Therefore, we did not provide the data repository information.”

Comment 4: Please include captions for your Supporting Information files at the end of your manuscript, and update any in-text citations to match accordingly. Please see our Supporting Information guidelines for more information: http://journals.plos.org/plosone/s/supporting-information.

Response 4: We have added captions at the end of the manuscript according to the supporting guidelines. In addition, we have updated the citations in the text. New additions are shown in red.

Page 37, Lines 59–596

Supporting information

S1 File. Dataset with all the participants. Re-measurement: result of a re-measurement by the same inspector > 6 months after the original measurement (30% of speakers with dysphoria).

Page 15, Lines 253–254

Details of all the participants, including healthy speakers, are shown in the S1 File.

Response to the editor

Comment: I have received 2 reviewer reports for the paper, “Relationships between maximum tongue pressure and second formant transition in speakers with different types of dysarthria”. Both reviewers note that the research question is of potential interest to a wide range of scholars, and that the methods and interpretation of results are generally sound. However, both reviewers expressed some concerns related to the details of how the F2 slope was calculated. Given that this is the major contribution of this paper, I agree that more details need to be included in your revisions about how F2 slope was calculated, particularly the rationale for the method chosen. For example, Reviewer 2 asks questions about whether the F2 slope was calculated across the entire word, or within each vowel separately, so more details on how the vowels in the target words were analyzed distinctly from the consonants in the words is needed. An additional figure, as suggested by reviewer 1, would be welcome. Reviewer 1 also suggests several places where the writing could be improved for clarity, and I suggest following this advice, particularly where redundancies can be avoided (also noted by Reviewer 2), or jargon can be removed or explained (particularly in the abstract). Reviewer 2 also suggests that some analysis be conducted related to the type and severity level of dysarthria, which would be welcome, given the title of the paper.

Response: We appreciate the reviewer for these constructive comments. We agree with all your suggestions; we have tried to reduce redundancy, particularly in the Abstract section. The authors’ responses to the comments are as follows:

Reviewer #1:

General comments:

This paper has a new important issue on speech evaluation for dysarthria patients. But the outside reader of your manuscript it could be a little hard to understand correctly. Could you look at the points below: 

Response: Thank you very much for your supportive comments and valuable suggestions for improving the quality of our manuscript.

Abstract

Comment 1: P3 L22-23

“The tongue pressure in speakers with dysarthria was significantly associated with the second formant transition slope.”

This sentence seems to have the same meaning as the one before this sentence. I recommend that you delete this sentence. 

Response 1: Thank you for your helpful suggestion. We have omitted the duplication. 

Comment 2: P3 L23-24

“This result suggests that the maximum isometric tongue strength is associated with articulation severity and tongue movement speed during articulation.”

The results showed that MTP was not correlated with speech intelligibility. The "the maximum isometric tongue strength is associated with articulation severity" in this sentence is inconsistent with the result. 

Response 2: Our data did not show that the articulation severity was related to MTP. Therefore, we have omitted "articulation severity and" from the sentence as indicated.

Page 3, Lines 37–38

This result suggests that the maximum isometric tongue strength is associated with tongue movement speed during articulation.

Introduction

Comment 3: P5. L56-57

“these are qualitative assessments”

Does this “these” mean "Elevation strength of the anterior tongue"? If so, the elevation strength of the anterior tongue is quantitative assessment, not qualitative assessment. 

Response 3: The word "these" meant "audibly articulatory precision and speech clarity." We have modified the sentence as follows:

Page 4, Lines 67–68 

However, since audibly articulatory precision and speech clarity are qualitative assessments, they affect the distribution of the data (ceiling or floor effect) [13].

Comment 4: P3 L57

“they affect the distribution of data (ceiling or floor effect).”

Please add references.

Response 4: Thank you for your kind suggestion. We have cited the reference for this sentence as follows:

Page 4, Lines 69 

“they affect the distribution of data (ceiling or floor effect) [13].”

Comment 5: P6 L67-69

“The hypothesis that … oral-DDK rates.”

Please add references.

Response 5: Thank you for your valuable suggestion. We have added the reference to this sentence:

Page 5, Lines 79–81

The hypothesis that the tongue strength affects the speech speed does not appear to be supported by the weak correlation between the tongue strength and the articulation and oral-DDK rates [13].

Material and Methods

Comment 6. P9 L121

“thickness, 0.5mm” 

I think it would be more appropriate to write the diameter rather than the thickness of the bite block. 

Response 1 : Thank you very much for your valuable suggestion. We have added the diameter of the bite block as follows:

Page 8, Lines 131–133

The TPM-01 comprises a disposable probe, an injection tube as a connector, and a hard ring (bite block; length, 8.5 mm; thickness, 0.5 mm; diameter, 6.0 mm) device (Fig 1). 

Comment 2. P12 L177-179

I suggest you should make a figure for analysis methods about onset time, offset time, movement duration, and movement extent. 

Response 2: Thank you very much for your valuable suggestion. We have created and included Figure 2 in our revised manuscript.

Page 13, Lines 202–207

Fig 2. A spectrogram of the word /taijo:/ produced by a man with stroke and mild unilateral upper motor neuron dysarthria. This figure illustrates a major two vowel sequence and semivowel F2 movements in /ai/ and /jo/. The black lines running through the estimated centers of the first and second formants (F1 and F2) illustrate the formant tracing. The F2 slope was determined or the time interval, Duration, and the corresponding F2 frequency change, Extent.

Comment 7. P12 L176-177

In the previous study, /aɪ/ was used to analyze the F2 movement. Why did you include /jo/ as well as /ai/ in the analysis in this study?

Response 7: Thank you very much for your pertinent comment. We apologize for not explaining this point accurately. 

There is no consensus on the existence of double vowels in Japanese. In a previous study of a limited number of native Japanese speakers of dysarthria, vowel-to-vowel sequences were used (e.g. Tamura et al, 2021); however, there were concerns regarding the measurement errors. On the other hand, semivowels are also present in Japanese, and since we could obtain formant trajectories relatively close to the double vowels, we decided to add not only the /ai/ vowel sequence but also the semivowel /j/, which has characteristics similar to double vowels, to increase the data and reduce the measurement error. In addition, we included /jo/, which is different from /ai/, owing to the possibility that the opening and closing of the mandible could affect the formants (Yunusova et al, 2012). We have added the following explanations to our manuscript, which also address Comment 8. 

Page 11, Lines 190–192

F2 movement was measured based on a previous report measuring diphthong /aɪ/ [33] and semivowel /jæ/ [22]. 

Page 11–12, Lines 197–200

 Based on a previous study of dysarthria [19], we averaged all the F2 slopes with different acoustic properties. This aimed to reduce the error from the actual syllable-specific tongue motion velocity, considering the possibility that mandibular opening and closing could be related to the context [22].

Comment 8. P12 L184-185

The F2 slope values of /ai/ and /jo/ were calculated as one average value in this study. Why did you calculate the two types of two-vowels (/ai/ and /jo/) with different acoustic characteristics together as one average value?

Response 8: We wanted to minimize the error in the measurements, as shown in Response 7; therefore, we included /ai/ and /jo/, in our analysis, which have different acoustic properties, and both show a rapid rise and fall in F2. By mixing the measurement targets, we thought we could adjust the error of each measurement value. Kindly refer to our previous response.

Results

Comment 9. P14 L223-224

This cross-sectional study population included speakers with dysarthria who were admitted to acute and convalescent hospitals between September 2017 and June 2020. Please describe how many participants were entered, and how many participants did not match the inclusion criteria and were excluded, resulting in 65 participants matching the inclusion criteria. In addition, please describe the reasons for the exclusion. 

Response 9: Thank you for your valuable suggestion. We have added the number of entries, the number of people excluded, and the reason for exclusion as follows:

Page 14, Lines 249–250

There were 72 entries in this study. However, seven speakers with severe cognitive impairments were excluded. 

Comment 10. P14-15 L225-227

“A significant difference in age was found between neurologically normal speakers and speakers with dysarthria (p < 0.001)”. 

This sentence only showed the difference in age between neurologically normal speakers and speakers with dysarthria. Please describe the statistically analyzed differences in sex between neurologically normal speakers and speakers with dysarthria. 

Response 10: Thank you for your helpful comment. The χ-square test showed a significant difference in the sex ratio between the groups. We have added the following sentence to the manuscript:

Page 12–13, Lines 216–217

The χ-square test was used to evaluate the differences in the sex ratio.

Page 14–15, Lines 252–253

A significant difference in the age and sex was found between neurologically normal speakers and speakers with dysarthria (age: p < 0.001, sex: p < 0.031)

Comment 11. P17 L244

“The average MTP of a speaker with dysarthria was 32.3 ± 9.9 kPa.” 

The median of MTP was shown in table 2, so this statement did not need to show the mean of MTP. 

Response 11: Thank you for pointing this out. We have omitted this sentence.

Discussion

Comment 12. P21 L303-304

“One of the factors behind these contradictory results is the inadequacy and imbalance of the subjects [10,13].” 

Please add a specific explanation for “inadequacy and imbalance of the subject”. 

Response 12: Thank you for your helpful suggestion. It has long been pointed out that dysarthria occurs in a variety of disease backgrounds (i.e., Parkinson's disease, ALS, stroke) and that different diseases produce different speech symptoms (Darley et al., 1969; Duffy 2020). In addition, various organs are involved in speech production, and speech characteristics may also differ depending on the damaged site. In order to correctly interpret the results of this study, it is necessary to collect and analyze a large number of cases for each of these diseases and disorders. Speech is reportedly affected the most by the severe loss of tongue muscle strength (Neel et al, 2015, Solomon et al, 2017). We have added these statements to the manuscript:

Page 23, Lines 358–362

　　　 　Specifically, depending on the threshold of the tongue muscle weakness affecting speech, the correlation may not be clear when only a small number of patients has the most severe muscle weakness [11,13]. In addition, there may be a difference in the degree of contribution to the speech impairment between diseases in which muscle weakness is the main symptom [4,10] and other diseases [3,13].

Comment 13. P21 L309-310

“Therefore, the results of this study support the weak correlation between the oral-DDK rate and tongue muscle strength.” 

In this study, there was no significant correlation between the oral-DDK rate and tongue muscle strength. I consider that this statement is over-interpreted. 

Response 13: Thank you for your suggestion. We agree with your opinion, and have modified the sentence as follows:

Page 23, Lines 366–367

Therefore, the results of this study do not support a strong relationship between MTP and the oral-DDK rate.

Comment 14. P21 L316-317

“This may have affected the results, as our study did not include participants with sustained combat injuries.” 

Please describe how the non-inclusion of the participants with sustained combat injurie affected the results in this study. 

Response 14: Thank you for your helpful suggestion. Solomon et al. (2017) included 16/55 people with combat injuries. A subgroup analysis with severely reduced tongue pressure also included 5/8 patients with brain injury as well as structural destruction of the orofacial region by blast explosion or gunshot wound and subnuclear paralysis. We believe that the orofacial disruption strengthens the relationship between the tongue muscle strength and speech function. Solomon et al. (2017) themselves mention a possible effect on the distribution in the combat injuries. Accordingly, we have added the following explanation:

Page 24, Lines 374–376

Combat injuries with orofacial injuries may have a greater impact on the tongue function necessary for speech. 

Comment 15. P22 L321

were included →　were included in this study.

Response 15: Thank you for your valuable suggestion. The sentence has been revised as follows:

Page 24, Line 377-379

Therefore, the difference in the correlation coefficient could not possibly detect a significant correlation because few participants with severe dysarthria were included in this study.

Comment 16. P23 L354-355

“Therefore, the smaller effect size of the difference between the F2 slope and MTP suggests that speakers with mild dysarthria have lower discriminative function.” 

This sentence is difficult to understand. Please describe why the smaller effect size of the difference between the F2 slope and MTP suggests that speakers with mild dysarthria have lower discriminative function. 

Response 16: Thank you for your valuable suggestion. To clarify our intentions, we have revised the text as follows:

Page 26–27, Lines 411–422

On the other hand, the effect size of the difference between the speakers with dysarthria and healthy participants on the MTP and F2 slope in this study was moderate. However, the effect of the difference in oral-DDK rate and speech intelligibility between healthy speakers and speakers with dysarthria was large. In a previous study, Japanese speakers with mild dysarthria (n = 16) showed no significant difference in the MTP compared with speakers without dysarthria (n = 29) [40]. In addition, the F2 slope was lower in speakers with moderate to severe dysarthria [21,41]. Thus, MTP and F2 slope are less capable in differentiating between healthy speakers and speakers with mild dysarthria than oral-DDK rate and speech intelligibility. Nevertheless, the findings of this study showed a moderate correlation between the F2 slope and MTP in speakers with dysarthria. The F2 slope is correlated with tongue muscle strength, which is stronger than the relationship between tongue muscle strength and other speech-related indicators.

Comment 17. P24 L357

between the F2 slope and MTP. →　between the F2 slope and MTP in speaker with dysarthria. 

Response 17: Thank you for your valuable suggestion. We have incorporated the following changes:

Page 26, Lines 419－421

Nevertheless, the findings of this study showed a moderate correlation between the F2 slope and MTP in speakers with dysarthria.

Comment 18. I consider that the oral-DDK rate and F2 slope are associated with the tongue movement speed during articulation. However, the MTP was significantly related to the F2 slope but not significantly related to the oral-DDK rate in this study. Please describe why the oral-DDK rate and F2 slope, which are associated with the tongue movement speed during articulation, had different results from each other. 

Response 18: Thank you for your helpful suggestion. F2 slope reflects the rate of back-and-forth movement of the tongue during articulation. The DDK rate, on the other hand, is affected by the on/off of the voice and the opening/closing of the mandible. Moreover, the DDK rate does not take into account the distortion of the target articulation. Therefore, we believe that the F2 slope better reflects the motor function of the tongue during articulation.

We have added an explanation in the discussion section as follows:

Page 27, Lines 423–434

In addition, the present study showed different results for the F2 slope and oral-DDK rate, which are related to tongue movement speed during articulation. This result is worthy of special mention. The DDK rate of /ta/ also depends on the speed of the voice on/off and mandibular opening/closing. In addition, the oral-DDK rate counts the number of syllables produced per second. Some articulation distortions (i.e., under shooting) are not reflected in the DDK rate measurements. Thus, both clear and unclear articulations are counted as one syllable (i.e., narrow or wide range of movement). The F2 slope, on the other hand, acoustically isolates the back-and-forth movement of the tongue during articulation and measures its speed. In a previous study, an oral-DDK task was not possible at maximum articulatory velocity [42]. Therefore, it is likely that the F2 slope better reflects the speed of the tongue movements than the oral-DDK rate, and this may have affected the results.

Reviewer #2

 This paper investigates the relationships between maximum tongue pressure and speech-related features, which include speech intelligibility, /ta/ DDK, and F2-slope. While speech intelligibility and /ta/ DDK were revealed to have no significant correlations with maximum tongue pressure, F2-slope was significantly correlated with the maximum tongue pressure (r=0.37, p<0.05). This paper proposes that the F2- slope may be useful at verifying the effect of tongue strength training. 

Response: Thank you very much for your supportive comments and suggestions for the improvement of the quality of our manuscript.

Comment 1: Is the manuscript technically sound, and do the data support the conclusions? 

- The idea of using F2-slope which can see both articulation accuracy and articulation rate, is convincing. 

- F2-slope is usually used for analyzing diphthongs(one vowel), and sometimes vowel sequences. However, I am concerned about this experiment design, which analyze the F2-slope in a word level. Are the features extracted from the start of /a/ and end of /o/? If this is the case, the design should be revised in a major manner. Especially, for word /gaito/, the consonant is in between the two vowels, which must interrupt in extracting the F2-slope. Even for /taijo/, the difference of F2 between /a/ and /i/ are much larger compared to difference of F2 between /a/ and /o/. Analyzing the F2-slope for each vowel sequence/semi-vowel may be more persuasive. 

Response 1: In this study, we did not analyze the whole word, but only the vowel-vowel transitions and semivowel (glide) portions. We analyzed three F2 slopes for each of these three locations: 1) the /ai/ transition in /taijo/, 2) the /jo/ glide in /taijo/, and 3) the /ai/ transition in /gaito:/. We have added the details of how we measured the F2 slope, as shown below, and created a measurement example in Figure 2.

Page 11, Lines 188–190

Details of the measurement targets: 1) /taijo/'s /ai/ transition, 2) /taijo/'s /jo/ glide, and 3) /gaito:/'s /ai/ transition. Figure 2 shows an example of the /taijo:/ measurement.

Page 11, Lines 195–197

Therefore, the F2 slopes are expressed in absolute values (Hz/ms) throughout this manuscript to eliminate the positive/negative sign caused by the target-inherent F2-movement direction.

Comment 2: Has the statistical analysis been performed appropriately and rigorously? 

Mann-Whitney U test is applied to investigate the difference between healthy speakers and dysarthric speakers. Pearson Correlation is used to examine the relationships between MTP results and speech function features. Subgroup analysis by gender are appropriately performed. 

Further subgroup analysis by dysarthric subtypes and severity levels should also be considered. In particular, the authors argue that the reason the study results do not agree with the previous results is because of the different distribution of speakers. Hence this analysis is necessary. 

Response 2: Thank you for your helpful suggestion. We performed additional analyses grouped by dysarthria subtypes and by maximum tongue pressure. For the subtypes, there were significant correlations between F2 slope and MTP for Mixed and Flaccid, with Flaccid having the strongest correlations with oral-DDK rate and speech intelligibility compared with the other subtypes. In the group with the lower maximum tongue pressure, all the speech indices were significantly correlated with maximum tongue pressure. 

We have added the following text to the Methods, Results, and Discussion section, respectively. Table 4 and Figure 4 have been added to the Results section.

Methods

Page 13, Lines 224–225

 In addition, a subgroup correlation analysis by the dysarthria subtype and MTP severity (divided into two groups by median) was performed.

Result

Page 20, Lines 311–318

　　　　　In the analysis by the dysarthria subtype, a significant correlation was observed between the MTP and F2 slope (Flaccid, rs = 0.786, p = 0.036; Mixed, rs = 0.640, p = 0.014; Table 4). Note that spastic (n = 3) and hyperkinetic (n = 1) types were excluded from the analysis due to their small sample size.

Table 4. Spearman’s rank correlation coefficients (two sided) between the maximum tongue pressure and speech measures for each subtype of speakers with dysarthria (Total n = 59)

Page 21–22, Lines 323–334

In the analysis by the groups categorized according to maximum tongue pressure, a significant correlation was observed between the MTP and all speech-related variables only in the lower MTP group (Speech intelligibility, rs = 0.397, p = 0.008; /ta/ DDK rate, rs = 0. 479, p = 0.038; F2 slope, rs = 0. 479, p = 0.038; Fig 4).

Fig 4. Bivariate scatter plot and best-fit regression line of the maximum tongue pressure and speech-related variables in the two speakers with dysarthria groups divided according to the maximum tongue pressure by median

Discussion

Page 25, Lines 388–390

　　　　　　The results of this study also showed that the correlation between the MTP and all the speech evaluations was stronger in the flaccid type, where muscle weakness was the main symptom, than in the other types.

Comment 3: Have the authors made all data underlying the findings in their manuscript fully available?

- The datasheet used for the statistical analysis is uploaded. All features used in the analysis are included

- However, each notation must be explained of its meaning. For example, what does POST and typeNo imply? 

Response 3: Thank you for your valuable suggestion. First, we changed "POST" to "Re-measurement" in the supporting information file. We have also added an explanation for " Re-measurement." In addition, I deleted "typeNo" since it was not relevant information for the readers.

Page 37, Lines 593–596

　　　　　S1 File. Dataset with all the participants. Re-measurement: Result of a re-measurement by the same inspector > 6 months after the original measurement (30% of speakers with dysarthria).

Comment 4: Further, each notations should be explained why each feature is included in the data. It is hard to understand why are height, weight, BMI, Albumin are included for the analysis. 

Response 4: Thank you for your valuable suggestion. The authors have reviewed the data and determined that the height, weight, BMI, and albumin do not need to be analyzed. Therefore, we have removed them from the data in the supporting information file.

Comment 5: Brief information of dysarthric speakers should be stated in the manuscript. (subtypes) 

Response 5: Thank you for your valuable suggestions. We have made the following additions:

Page 16, Lines 266–268

The breakdown of the dysarthria subtypes in the cases was as follows: spastic, 3; flaccid, 7; hypokinetic, 6; hyperkinetic, 1; ataxic, 7; unilateral upper motor neuron, 16; mixed, 14; and undetermined, 9.

Comment 6: 4. Is the manuscript presented in an intelligible fashion and written in standard English?

- The manuscript is presented in an intelligible fashion and well-written in standard English. 

- Though there are some redundant sentences that should be deleted for the final submissions. 

Abstract is quite confusing, especially the last sentence : "However, based on the degree of these correlations, the hypothesis that the relationship between the maximum force of the tongue and speech function is weak is also strengthened." This sentence blurs the main idea of this paper, which suggests the significant correlation between maximum force of the tongue and F2-slope. 

Response 6: Thank you for pointing this out. We have removed the aforementioned sentence and modified the entire abstract to correct the redundancy.

Comment 7: line 338 : F2-slope -> F2 (second formant) 

Response 7: Thank you for your valuable suggestion. We have split the sentence into two and modified it as follows:

Page 26, Lines 399–402

 F2 roughly corresponds to the back-and-forth movement of the tongue [15]. The F2 slope, calculated from the movement duration and extent of the F2, is an acoustic index that correlates with the perceived accuracy of vowels [16].

Comment 8: line 340 : It -> F2-Slope

Response 8: Thank you for your valuable suggestion. We have corrected it accordingly.

Page 25–26, Lines 402–404

The F2 slope is speculated to reflect the speed of the tongue movement during articulation. In this study, the correlation between the F2 slope and tongue muscle strength was similar to all speakers with dysarthria in the sex-separated subgroups.

---

## [Decision Letter · Decision Letter 1]

10 Feb 2022

PONE-D-21-32058R1Relationships between maximum tongue pressure and second formant transition in speakers with different types of dysarthriaPLOS ONE

Dear Dr. Tamura,

Thank you for submitting your manuscript to PLOS ONE. After careful consideration, we feel that it has merit but does not fully meet PLOS ONE’s publication criteria as it currently stands. Therefore, we invite you to submit a revised version of the manuscript that addresses the points raised during the review process.

We look forward to receiving your revised manuscript.

Kind regards,

Sara Finley, Ph.D.

Academic Editor

PLOS ONE

Journal Requirements:

Additional Editor Comments (if provided):

Both original reviewers have read the revised manuscript and believe that their concerns have been met in the revision. However, both reviewers have minor suggestions for clarification and wording. These are important in order to be sufficiently clear for a broad audience. I also have some minor comments related to clarity and wording:

Line 120: "Comprised 20 participants" should be 'comprised of 20 participants"

Line 130: "highly guaranteed" seems somewhat superlative. I would suggest toning this down, or noting that the measurement is believed to be reproducible and reliable

Lines 206-207- "The F2 was determined or the..." This sentence is somewhat confusing to read. Please make sure that capitalization is consistent.

Line 319: "Bold value indicates and" This sentence is confusing. It seems that a word is missing here. Value should be 'values'?

Line 333. Make sure to have consistent capitalization for 'Lower'.

Reviewers' comments:

Reviewer's Responses to Questions

**Comments to the Author**

1. If the authors have adequately addressed your comments raised in a previous round of review and you feel that this manuscript is now acceptable for publication, you may indicate that here to bypass the “Comments to the Author” section, enter your conflict of interest statement in the “Confidential to Editor” section, and submit your "Accept" recommendation.

Reviewer #1: (No Response)

Reviewer #2: All comments have been addressed

2. Is the manuscript technically sound, and do the data support the conclusions?

Reviewer #1: Yes

Reviewer #2: Yes

3. Has the statistical analysis been performed appropriately and rigorously? 

Reviewer #1: Yes

Reviewer #2: Yes

4. Have the authors made all data underlying the findings in their manuscript fully available?

Reviewer #1: Yes

Reviewer #2: Yes

5. Is the manuscript presented in an intelligible fashion and written in standard English?

Reviewer #1: Yes

Reviewer #2: Yes

6. Review Comments to the Author

Reviewer #1: Relationships between maximum tongue pressure and second formant transition in　speakers with different types of dysarthria

General comments:

This paper has been properly revised from the first manuscript. But a part of your manuscript remain a little hard to understand correctly. Could you look at the point below:

Material and Methods

1. P12 L202-207

“Fig 2. A spectrogram of the word…. corresponding F2 frequency change, Extent.”

In Figure 2, the analysis methods of F2 onset and F2 offset for /ai/ and /jo/ were unclear. Please explain the detailed definition of F2 onset and F2 offset for /ai/ and /jo/.

2. P12 L206-207

The F2 slope was determined or the time interval, Duration, and the corresponding F2 frequency change, Extent.

→ The F2 slope was determined or the time interval, duration, and the corresponding F2 frequency change, extent.

Reviewer #2: The authors have adequately addressed the given comments raised in a previous round of review. I feel confident that the manuscript has become much clearer. The manuscript is technically sound, and the data support the conclusions. The statistical analysis has been performed appropriately and rigorously (Mann-Whitney U test - assessment of the difference between dysarthric speakers vs healthy speakers; Pearson correlation - correlations between MTP and speech-related features (speech intelligibility, /ta/ DDK, F2-slope) within speakers with dysarthria. However, there are some minor comments for the improvements of clarity.

# Abstract

line 31: Some speakers with dysarthria -> Speakers with dysarthria

- The paper reports the significant difference between dysarthric speakers and healthy speakers. To state as 'some speakers' seems unnecessary.

line 38: This result suggests that the maximum isometric tongue strength is associated with tongue movement speed during articulation.

- oral DDK is also known to represent the tongue movement speed during articulation. Elaborating on the difference between oral DDK and F2-slope may help the readers better understand the paper's main contribution.

# Introduction

line 49-50: does speech clarity align with speech intelligibility? Please elaborate on this terminology. It is difficult to relate 'severity' with speech-related indicators (compared to articulation rate, oral-DDK rate). Please elaborate.

line 60: overall severity (of dysarthria? muscle strength?)

line 62: what does /ta/<5.8 syllables/s infer? Stating the oral-DDk rate for healthy speakers or mild dysarthria is necessary.

line 63, line 67: please explain what speech audibility is. The authors are using 3 terminologies in the manuscript - speech intelligibility, speech clarity, speech audibility.

line 71 : F2 slope -> F2 (second formant)

line 77-78: relatively slow changes in tongue shape. Please elaborate. How is tongue shape related to the decrease in the F2-slope? In addition, tongue shape seems not to be related to movement speed, which is the essence of this paper.

# Materials and Methods - Participants

- Information of dysarthric speakers (how many speakers, Gender, Age, etc) is omitted.

line 115: please account for the reason why height, weight, albumin, BMI are collected for this study.

# Materials and Methods - Oral-DDK rate

line 173: 3s of the central parts: Please explain why the authors decided to use only the central parts of the audio, rather than the full audio.

line 192: Specific explanation of how the authors determined the onset and offset of F2 is needed for replication.

# Materials and Methods - Second formant transition

fig 2 - fast formant > first formant

# Results

line 301-302 : Correlation between MTP & speech intelligiblity, MTP & /ta/ DDK by sex groups should also be presented by gender.

line 311- 314: Descriptions for the analysis of other subtypes are missing. (hypokinetic, ataxic, etc.)

# Discussion

line 337-347: The fact that the authors analyzed the correlations between MTP and speech-related indicators within dysarthric speakers does not stand out in this paragraph.

line 388-390: According to the results, speech intelligibility and oral-DDK measurements did not show significant correlations with MTP.

line 407: some speakers with dysarthria - please concretely describe the corresponding speakers.

# Conclusion

line 458: the appropriateness of articulation and speed

7. PLOS authors have the option to publish the peer review history of their article (what does this mean?). If published, this will include your full peer review and any attached files.

Reviewer #1: No

Reviewer #2: No

---

## [Author Response · Author response to Decision Letter 1]

18 Feb 2022

Point-by-point responses to comments from the Editor and Reviewers

Thank you for reviewing our work. We appreciate all your comments and suggestions. We have revised the manuscript accordingly. Our point-by-point responses are presented below.

Response to Journal Requirements

Comment 1: Please review your reference list to ensure that it is complete and correct. If you have cited papers that have been retracted, please include the rationale for doing so in the manuscript text, or remove these references and replace them with relevant current references. Any changes to the reference list should be mentioned in the rebuttal letter that accompanies your revised manuscript. If you need to cite a retracted article, indicate the article’s retracted status in the References list and also include a citation and full reference for the retraction notice.

Response 1: Thank you for your suggestion. The doi of references 19 and 41 in the reference list were incomplete and have been corrected. The retracted papers are not cited in this paper. Other minor corrections have also been made.

Response to the editor

General comments:

Both original reviewers have read the revised manuscript and believe that their concerns have been met in the revision. However, both reviewers have minor suggestions for clarification and wording. These are important in order to be sufficiently clear for a broad audience. I also have some minor comments related to clarity and wording:.

Response: We appreciate the reviewer for their constructive comments. The authors’ responses to the comments are as follows:

Comment 1: Line 129: "Comprised 20 participants" should be 'comprised of 20 participants"

Response 1: Thank you for the detailed confirmation. Our manuscript was submitted for professional English editing and proofreading, and we were informed that “comprised of” is a grammatically incorrect phrase. We hope that retaining the original phrase here will not be an issue..

Taken from the Merriam-Webster website:

“Although comprised of is an established standard for "being composed or constituted of," it is often liable to criticism and scrutiny. The correct version put forward by grammar guides is to use "composed of" or "comprises" such as "the cake is composed of flour and eggs" or "comprises flour and eggs."

Comment 2: Line 139: "highly guaranteed" seems somewhat superlative. I would suggest toning this down, or noting that the measurement is believed to be reproducible and reliable

Response 2: Thank you for your helpful suggestion. Accordingly, we have made the following revision in the sentence:

Page 8, Line 139

The reproducibility and reliability of this device has been validated in a previous study [26].

Comment 3: Lines 225-226: "The F2 was determined or the..." This sentence is somewhat confusing to read. Please make sure that capitalization is consistent.

Response 3: Thank you for your helpful suggestion. We have made the following revision in the sentence:

Page 13, Lines 225-226

Duration is the time interval of the F2 movement. Extent is the range of frequency change in the F2 movement.

Comment 4: Line 344: "Bold value indicates and" This sentence is confusing. It seems that a word is missing here. Value should be 'values'?

Response 4: Thank you for your helpful suggestion. We have made the following revisions in the text:

Page 22, Line 344: Values in bold and with asterisks indicate that rs is significant (p < 0.05) (two-tailed).

Comment 5: Line 357: Make sure to have consistent capitalization for 'Lower'.

Response 5: Thank you for your helpful suggestion. Accordingly, we have removed the capitalization from the sentence:

Page 23, Lines 357-359

　　　　Speakers with lower maximum tongue pressure and dysarthria (lower group) tended to have lower speech-related variables, such as lower maximum tongue pressure.

Reviewer #1:

General comments:

This paper has been properly revised from the first manuscript. But a part of your manuscript remain a little hard to understand correctly. Could you look at the point below:

Response: Thank you for your supportive comments and valuable suggestions for improving the quality of our manuscript. The authors’ responses to the comments are as follows:

Material and Methods

Comment 1: P13 L221-225

“Fig 2. A spectrogram of the word…. corresponding F2 frequency change, Extent.”

In Figure 2, the analysis methods of F2 onset and F2 offset for /ai/ and /jo/ were unclear. Please explain the detailed definition of F2 onset and F2 offset for /ai/ and /jo/. 

Response 1: Thank you for comment. We have adjusted the sound spectrogram in Fig. 2 to improve clarity regarding the whole picture. We have also made the following revisions in the text:

Page 12, Lines 205–212

The F2 linear predictive coding tracks on a wide-band spectrogram (analysis bandwidth, 300 Hz) were manually edited and identified. The 20/20 rule (specifying a frequency change of ≥20 Hz during 20 ms in the transition onset and offset) was applied [41]. If the F2 track was unclear due to hoarseness or other noises, we identified it by referring to the changes in F1 (opening and closing of the mandible) that were almost synchronized with F2 in the /ai/ and /jo/ sequences. Thus, relatively stationary portions of the vowel before or after the target transition were not included in the analysis.

Page 13, Lines 226–228

The F2 movement for each syllable is determined by a change in frequency of at least 20 Hz during a 20 ms period, not including the relatively stationary portions before and after the syllable.

Comment 2: P12 L225-226

The F2 slope was determined or the time interval, Duration, and the corresponding F2 frequency change, Extent.

→ The F2 slope was determined or the time interval, duration, and the corresponding F2 frequency change, extent.

Response 2: Thank you for your helpful suggestion. We have revised this for more clarity.

Page 13, Lines 225-226

Duration is the time interval of the F2 movement. Extent is the range of frequency change in the F2 movement.

Reviewer #2

General comments:

The authors have adequately addressed the given comments raised in a previous round of review. I feel confident that the manuscript has become much clearer. The manuscript is technically sound, and the data support the conclusions. The statistical analysis has been performed appropriately and rigorously (Mann-Whitney U test - assessment of the difference between dysarthric speakers vs healthy speakers; Pearson correlation - correlations between MTP and speech-related features (speech intelligibility, /ta/ DDK, F2-slope) within speakers with dysarthria. However, there are some minor comments for the improvements of clarity. 

Response: Thank you for your supportive comments and suggestions for the improvement of the quality of our manuscript. The authors’ responses to the comments are as follows:

# Abstract

Comment 1: line 31: Some speakers with dysarthria -> Speakers with dysarthria

- The paper reports the significant difference between dysarthric speakers and healthy speakers. To state as 'some speakers' seems unnecessary.

Response 1: Thank you for your helpful suggestion. We have incorporated your suggestion in our paper.

Comment 2: line 39: This result suggests that the maximum isometric tongue strength is associated with tongue movement speed during articulation.

- oral DDK is also known to represent the tongue movement speed during articulation. Elaborating on the difference between oral DDK and F2-slope may help the readers better understand the paper's main contribution. 

Response 2: Thank you for your helpful suggestion. We have made the following revision in the sentence:

Page 3, Lines 37–39

The oral diadochokinesis rate, which is related to the speed of articulation, is affected by voice on/off, mandibular opening/closing, and range of motion. In contrast, the second formant slope was less affected by these factors.

# Introduction

Comment 3: line 51-53: does speech clarity align with speech intelligibility? Please elaborate on this terminology. It is difficult to relate 'severity' with speech-related indicators (compared to articulation rate, oral-DDK rate). Please elaborate. 

Response 3: Thank you for your question. 

-Yes, speech clarity is the same as speech intelligibility. For more clarity, the term “speech intelligibility” has been used consistently in the paper. In addition, “severity” has been deleted because it includes impressions other than speech. We have made the following revision in the sentence:

Page 3, Lines 51–53

However, current reviews have reported no significant relationship between tongue strength and speech-related indicators, such as speech intelligibility, articulation rate, and oral-DDK rate [6–8].

Comment 4: line 63: overall severity (of dysarthria? muscle strength?) 

Response 4: Thank you for your comment. Overall severity is an auditory measure of dysarthria, and corresponds to a comprehensive assessment of speech intelligibility and naturalness. We have added the following to the sentence to make it clearer:

Page 4, Lines 61–64

Additionally, speakers with dysarthria in whom tongue muscle strength is lower than the lower limit of normal speakers have moderate-to-severely reduced articulatory precision and overall severity (including speech intelligibility and naturalness) [13].

Comment 5: line 65: what does /ta/<5.8 syllables/s infer? Stating the oral-DDK rate for healthy speakers or mild dysarthria is necessary.

Response 5: Thank you for your valuable suggestion. We have made the following additions:

Page 4, Lines 66–67

In a previous cross-sectional study [13], dysarthria speakers (n = 8) with severe anterior tongue elevation muscle strength had an oral-DDK rate of <5.8 syllable/s for the syllable /tʌ/. In contrast, 44.6% of the remaining speakers with dysarthria had an oral-DDK rate of >5.8 syllables/s.

Comment 6: line 68, line 72: please explain what speech audibility is. The authors are using 3 terminologies in the manuscript - speech intelligibility, speech clarity, speech audibility.

Response 6: Thank you for pointing this out. We were using these terms in the same context. Therefore, we have unified all three terms into “speech intelligibility.”

Page 3, Lines 53; Page 4, Lines 68, 72; Page 5, Lines 73

Comment 7: line 76 : F2 slope -> F2 (second formant)

Response 7: Thank you for the detailed confirmation. We have defined the second formant as F2 in the sentence preceding this one(line 74). We hope that retaining the original phrase here will not be an issue..

Comment 8: line 82-83: relatively slow changes in tongue shape. Please elaborate. How is tongue shape related to the decrease in the F2-slope? In addition, tongue shape seems not to be related to movement speed, which is the essence of this paper. 

Response 8: Thank you for your comment. The tongue is a muscle without any joints. Therefore, “relatively slow changes in tongue shape” has the same meaning as “slow movement of the tongue.” In other words, the back-and-forth motion of the tongue during articulation is slower, resulting in a smaller F2 slope. To make it clearer, we have added the following sentence.

Page 5, Lines 81–85

The clear explanation for the decrease in the F2 slope in speakers with dysarthria is the relatively slow changes in tongue shape [22]. Specifically, the back-and-forth motion of the tongue during articulation is slower and/or the range of movement is narrower, resulting in a longer and thinner change in the F2 movement.

# Materials and Methods - Participants

Comment 9: Information of dysarthric speakers (how many speakers, Gender, Age, etc) is omitted.

line 120: please account for the reason why height, weight, albumin, BMI are collected for this study. 

Response 9: Thank you for your valuable suggestion. These factors were considered to account for the possible effects on tongue muscle strength and speech caused by factors other than the primary disease causing dysarthria. The following sentences have been added to the manuscript.

Page 7, Lines 122–124

　　　　These factors were considered to account for the possible effects on tongue muscle strength and speech caused by factors other than the primary disease causing dysarthria.

# Materials and Methods - Oral-DDK rate

Comment 10: line 183: 3s of the central parts: Please explain why the authors decided to use only the central parts of the audio, rather than the full audio. 

Response 10: Thank you for your valuable suggestion. We used a 3-second duration of the middle part of the audio to reduce the effects of speech irregularities during speech onset and respiratory dysfunction during the second half of the task. In the syllable repetition task, the first syllable is often uttered longer than the following syllable in normal subjects (Ackerman et al, 1995). In speakers with dysarthria, it is possible that the initial effect is due to a freezing at movement onset (e.g., hypokinetic type). In addition, speakers with dysarthria are more likely to be affected by breath-holding in the second half of oral-DDK due to reduced respiratory function. The following sentences have been added to the manuscript.

Page 11, Lines 183–186

　　　　To reduce the effects of speech irregularities such as freezing, slurring, or syllable prolongation during speech onset, or respiratory dysfunction during the second half of the task, we extracted ~3 s of recorded data from the middle parts of the audio for the analysis.

Comment 11: line 204: Specific explanation of how the authors determined the onset and offset of F2 is needed for replication. 

Response 11: Thank you for your comment. Based on your suggestion, I added the method of determining the onset and offset.

Page 12, Lines 205–212

The F2 linear predictive coding tracks on a wide-band spectrogram (analysis bandwidth, 300 Hz) were manually edited and identified. The 20/20 rule (specifying a frequency change of ≥20 Hz during 20 ms in the transition onset and offset) was applied [34]. If the F2 track was unclear due to hoarseness or other noises, we identified it by referring to the changes in F1 (opening and closing of the mandible) that were almost synchronized with F2 in the /ai/ and /jo/ sequences. Thus, relatively stationary portions of the vowel before or after the target transition were not included in the analysis.

# Materials and Methods - Second formant transition

Comment 12: fig 2 - fast formant > first formant

Response 12: Thank you for pointing this out. We have corrected this.

# Results

Comment 13: line 322-323 : Correlation between MTP & speech intelligibility, MTP & /ta/ DDK by sex groups should also be presented by gender.

Response 13: Thank you for your valuable suggestion. We have added the following sentence:

Page 20, Lines 324-326

The correlation between MTP and speech intelligibility was significant only in males (rs = -0. 328, p = 0.030). There was no significant correlation between MTP and the /ta/ DDK rate in either sex.

Comment 14: line 335- 337: Descriptions for the analysis of other subtypes are missing. (hypokinetic, ataxic, etc.) 

Response 14: Thank you for pointing this out. We have added the following sentence:

Page 21, Lines 337–338

There was no significant correlation between any of the combinations in the other subtypes.

# Discussion

Comment 15: line 362-373: The fact that the authors analyzed the correlations between MTP and speech-related indicators within dysarthric speakers does not stand out in this paragraph.

Response 15: We have added the following sentence:

Page 23, Lines 363–364

In this study, we measured the tongue pressure in speakers with various types of dysarthria, following which, we conducted a correlation analysis between MTP and speech-related indicators..

Comment 16: line 414–417: According to the results, speech intelligibility and oral-DDK measurements did not show significant correlations with MTP. 

Response 16: Thank you for pointing this out. We have made the following revision:

Page 26, Lines 414–417

　　　　In our results, speech intelligibility and oral-DDK rate were not significantly correlated with MTP in all subtypes. However, the flaccid type (n = 7) had the highest correlation coefficient among all types (speech intelligibility, rs = -0.436; /ta/ DDK rate, rs = 0.517). Additional subtype-specific studies are warranted.

Comment 17: line 432: some speakers with dysarthria - please concretely describe the corresponding speakers.

Response 17: Thank you for your valuable suggestion. We have reconsidered the results of this study and believe that it is not possible to determine the specific characteristics of the speakers. Therefore, we have deleted “some” and modified the sentence as follows:

Page 27, Lines 432–435

In summary, in speakers with dysarthria, tongue strength may be associated with tongue movement velocity during articulation. Our findings partially support the hypothesis that “muscle weakness is associated with slow speech” [2].

# Conclusion

Comment 18: line 483: the appropriateness of articulation and speed

Response 18: Thank you for your suggestion. We have added this to the text.

---

## [Editor Report · Decision Letter 2]

22 Feb 2022

Relationships between maximum tongue pressure and second formant transition in speakers with different types of dysarthria

PONE-D-21-32058R2

Dear Dr. Tamura,

We’re pleased to inform you that your manuscript has been judged scientifically suitable for publication and will be formally accepted for publication once it meets all outstanding technical requirements.

Kind regards,

Sara Finley, Ph.D.

Academic Editor

PLOS ONE

Additional Editor Comments (optional):

Thank you for carefully revising and considering each comment from the editors and reviewers. I believe that this revision has successfully addressed all concerns and the paper should be published.

Reviewers' comments:

N/A

---

## [Editor Report · Acceptance letter]

24 Feb 2022

PONE-D-21-32058R2 

Relationships between maximum tongue pressure and second formant transition in speakers with different types of dysarthria 

Dear Dr. Tamura:

I'm pleased to inform you that your manuscript has been deemed suitable for publication in PLOS ONE. Congratulations! Your manuscript is now with our production department. 

Kind regards, 

on behalf of

Dr. Sara Finley 

Academic Editor

PLOS ONE